# Contrast Agents during Pregnancy: Pros and Cons When Really Needed

**DOI:** 10.3390/ijerph192416699

**Published:** 2022-12-12

**Authors:** Federica Perelli, Irene Turrini, Maria Gabriella Giorgi, Irene Renda, Annalisa Vidiri, Gianluca Straface, Elisa Scatena, Marco D’Indinosante, Laura Marchi, Marco Giusti, Antonio Oliva, Simone Grassi, Carmen De Luca, Francesco Catania, Giuseppe Vizzielli, Stefano Restaino, Giuseppe Gullo, Georgios Eleftheriou, Alberto Mattei, Fabrizio Signore, Antonio Lanzone, Giovanni Scambia, Anna Franca Cavaliere

**Affiliations:** 1Azienda USL Toscana Centro, Gynecology and Obstetric Department, Santa Maria Annunziata Hospital, 50012 Florence, Italy; 2Azienda USL Toscana Centro, Gynecology and Obstetric Department, Santo Stefano Hospital, 59100 Prato, Italy; 3Division of Obstetrics and Gynecology, Department of Biomedical, Experimental and Clinical Sciences, University of Florence, 50134 Florence, Italy; 4School of Medicine, Catholic University of the Sacred Hearth, 00168 Rome, Italy; 5Obstetrics and Gynecology Unit, Policlinico Abano Terme, 35031 Abano Terme, Italy; 6Department of Health Surveillance and Bioethics, Section of Legal Medicine, Fondazione Policlinico A. Gemelli IRCCS, Università Cattolica del Sacro Cuore, 00168 Rome, Italy; 7Teratology Information Service, Fondazione Policlinico Universitario Agostino Gemelli IRCCS, 00168 Rome, Italy; 8Department of Obstetrics and Gynecology, Ospedale “Santa Maria Alla Gruccia”, 52025 Montevarchi, Italy; 9Department of Medicinal Area (DAME) Clinic of Obstetrics and Gynecology, Santa Maria della Misericordia Hospital, Azienda Sanitaria Universitaria Friuli Centrale, 33100 Udine, Italy; 10IVF Public Center, Azienda Ospedaliera Ospedali Riuniti (AOOR) Villa Sofia Cervello, University of Palermo, 90146 Palermo, Italy; 11Poison Control Center and Teratology Information Service, Hospital Papa Giovanni XIII, 24127 Bergamo, Italy; 12Obstetrics and Gynecology Unit, Santo Eugenio Hospital, 00144 Rome, Italy; 13School of Medicine, Unicamillus University Rome, 00131 Rome, Italy; 14Department of Woman and Child Health and Public Health, Fondazione Policlinico Universitario Agostino Gemelli IRCCS, 00168 Rome, Italy; 15Division of Gynecologic Oncology, Fondazione Policlinico Universitario Agostino Gemelli IRCCS, 00168 Rome, Italy; 16Division of Gynecology and Obstetrics Fatebenefratelli Isola Tiberina, 00186 Rome, Italy

**Keywords:** contrast media, radiodiagnostic, pregnant, magnetic resonance imaging, computerized tomography, positron emission tomography, ultrasound, ionizing radiation

## Abstract

Many clinical conditions require radiological diagnostic exams based on the emission of different kinds of energy and the use of contrast agents, such as computerized tomography (CT), positron emission tomography (PET), magnetic resonance (MR), ultrasound (US), and X-ray imaging. Pregnant patients who should be submitted for diagnostic examinations with contrast agents represent a group of patients with whom it is necessary to consider both maternal and fetal effects. Radiological examinations use different types of contrast media, the most used and studied are represented by iodinate contrast agents, gadolinium, fluorodeoxyglucose, gastrographin, bariumsulfate, and nanobubbles used in contrast-enhanced ultrasound (CEUS). The present paper reports the available data about each contrast agent and its effect related to the mother and fetus. This review aims to clarify the clinical practices to follow in cases where a radiodiagnostic examination with a contrast medium is indicated to be performed on a pregnant patient.

## 1. Introduction

The use of imaging methods with or without contrast agent administration in pregnant and breastfeeding women has been increasing in the past decades [1]. In this subset of patients, it is mandatory to consider both maternal and fetal effects deriving from the use of a contrast media. The use of contrast media is generally avoided by physicians and patients, mainly because of the lack of wide prospective studies on humans concerning this issue and because the effects on a human embryo and fetus are not completely known. In our opinion, it is important to be aware of the high value of such diagnostic methods in specific cases and of the actual low incidence of adverse effects during pregnancy if properly used, following the International Associations’ guidelines, in order to guarantee the safe and effective use of such instruments.

We have reviewed the current literature and guidelines dealing with the principal contrast agents employed in clinical practice, explaining the risks of administration for the mother and the fetus, the adverse effects associated, and the main clinical recommendations for their use.

## 2. Classification of Contrast Agents

Contrast agents can be defined as any substance introduced into the body during imaging examinations, to improve the visualization and detection rate of internal structures. Medical imaging includes methods that use ionizing and non-ionizing radiation. Ionizing radiation consists of either particulate or electromagnetic energy such as alpha particles and beta particles, which have moderate penetrating power and can ionize tissues changing their normal structure and causing two types of effects: deterministic and stochastic [2]. Deterministic effects involve the loss of tissue function. If the radiation dose is distributed over time, the cellular repair mechanisms permit the tissue to recover from the damage. This implies a threshold dose above which the tissue will exhibit permanent damage since the radiation dose exceeds the capabilities of innate cellular repair mechanisms. Pregnancy loss, fetal malformations, neurodevelopment abnormalities, and fetal growth retardation, have a deterministic effect whereby a threshold or No-Adverse-Effect Level (NOAEL) exists [3]. On the other hand, stochastic effects have no threshold dose and can occur at any radiation dose. Nonionizing radiation (such as ultrasound waves, visible light, microwaves, and magnetic resonance imaging) “has enough energy to move around atoms in a molecule or cause them to vibrate, but not enough to remove electrons”. A fetus is partly protected from radiation effects by the mother’s surrounding soft tissues and uterus, both of which could stop alpha and beta particles from penetration. Nevertheless, if alpha and beta particles are ingested, injected, or inhaled severe adverse effects on the fetus could develop: radioactive material could accumulate in the pregnant woman’s bladder, causing internal radiation exposure [3]. Conversely, gamma and x-rays directed toward the abdomen of a pregnant woman not appropriately shielded can reach the fetus.

Medical imaging based on ionizing radiation includes the following: computerized tomography (CT), positron emission tomography (PET), and X-ray imaging.

Non-ionizing imaging is represented by magnetic resonance imaging (MRI) and ultrasound (US).

Iodinated and gadolinium-based contrast agents (GBCA) are the most frequently used, as CT and MRI, respectively, are the imaging modalities mainly used in daily practice [4,5].

### 2.1. Iodinate Contrast Agents

CT represents the imaging modality used the most often in clinical practice, especially chest CT, as trauma and suspected pulmonary emboli represent the principal indications for imaging examinations [1]. CT employs ionizing electromagnetic radiations (X-rays) to create cross-sectional slice pictures inside selected areas of the body from different angles, creating a three-dimensional picture.

Ionizing radiation, including X-ray, has moderate penetrating power and can ionize tissues, changing their normal structure and causing two types of effects: deterministic and stochastic [3].

There is evidence that shows that some adverse effects could occur at lower doses, too. As the Radiation Effects Research Foundation has demonstrated, radiation exposure during organogenesis and fetal development may lead to neurobehavioral changes, adverse effects on the central nervous system, malformations, low birth weight, and growth restriction [6].

Iodinate agents are used in CT to improve the detection rate of imaging.

The Food and Drug Administration (FDA) classifies the iodinated contrast agents as pregnancy category B drugs as they are considered safe for pregnant women and lactating mothers, except for diatrizoate meglumine and diatrizoate meglumine sodium, which are classified as category C drugs: “Animal reproduction studies have shown an adverse effect on the fetus and there are no adequate and well-controlled studies in human beings, but potential benefits may justify the use of the drug in pregnant women despite the potential risks” [7]. The American College of Obstetricians and Gynecologists recommends their use in cases of effective need for additional diagnostic information which could affect maternal or fetal care and outcome [8].

Iodinate contrast media can be classified according to osmolarity, ionicity, and the number of benzene rings [9]. Ionic iodinate contrast media have one benzene ring monomer containing three iodine atoms and a side chain with a carboxylic acid (–COOH) group [9]. Their osmolality is 5–7 times that of normal serum, so they have been classified as hypertonic and high-osmolar iodinate contrast media. Non-ionic iodinate contrast media have one benzene ring monomer with various side chains containing polar alcohol (–OH) groups, but no –COOH groups. Due to its non-ionic characteristics, the osmolality is decreased to 2–3 times that of normal serum, but its radiopacity remains similar. Compared to the osmolality of ionic iodinate contrast media, monomeric non-ionic iodinate contrast media are classified as hypotonic or low-osmolar iodinate contrast media. The incidence of mild and moderate contrast reactions is higher for high-osmolar contrast media (6–8%) than for low-osmolar contrast media (0.2%). Nonionic contrast agents, characterized by a lower osmolality, cause less adverse reactions compared to ionic agents, therefore the formers have mostly been used in X-ray-related studies in recent years [10].

Adverse effects in pregnant women are the same as those of the general population: the most common being hypersensitivity, thyroid dysfunction, and nephropathy. There is no evidence regarding teratogenic effects in humans, but studies on animals suggest no teratogenic or mutagenic effects if used during pregnancy [9]. The transplacental passage of ionic agents has been demonstrated by experimental studies on animals: after entering the fetal bloodstream, they are excreted in fetal urine in the amniotic fluid and then swallowed by the fetus. There is no clear evidence of the transplacental passage of nonionic agents.

Cohort studies suggest that exposure to iodinate contrast agents during pregnancy may cause thyroid dysfunction in offspring, including transient hypothyroidism and goiter. The fetal thyroid starts to become active and produces hormones from the beginning of the second trimester, with an increase in iodine uptake. It is highly sensitive to fluctuations in maternal iodine concentration in this period [11]. Actually, the major side effects are reported when ionic contrast agents are administrated during pregnancy and even in the preconceptional phase: the use of liposoluble ionic agents in hysterosalpingography in infertile women seems to correlate with a higher risk of thyroid dysfunction (2.4% vs. 0.7% in non-exposed fetuses) [12]. Contrarily, nonionic iodine media agents do not affect TSH and T4 levels in offspring, as their presence in fetal blood is transient [13].

The European Society of Urogenital Radiology recommended that neonatal thyroid function should be checked during the 1st week after birth if iodinated contrast media was given during pregnancy [14].

Iopamidol (Isovue) is a nonionic, low osmolarity, monomeric iodine agent which passes the placenta barrier. It has been demonstrated to be safe for fetal thyroid function, reproductive outcomes, and teratogenic effects in animals [15,16]. Iopromide (Ultravist) is the most used iodine contrast agent in CT: it is a nonionic, low osmolarity, and hydrosoluble agent with a transplacental passage. It is safe for teratogenic and mutagenic effects. One case report noted a transient TSH level increase in newborns when exposed during pregnancy, without thyroid hormone dysregulation [13].

The safety of such media agents is widely evident. Nevertheless, the lack of clinical studies on humans favors the doubts about their use in pregnant women, as this has been described only in a few case reports.

As far as concerns breastfeeding, the iodinated contrast agents are completely cleared from the mother’s bloodstream in 24 h and their half-life in blood is 2 h. Less than 1% of the administrated iodine agent is excreted in milk and less than 1% of it is assumed by the breastfed baby [9]. Consequently, the median dose absorbed by the baby is 0.05% of the recommended safe dose to administrate if the infant needs a diagnostic imaging examination [17].

In conclusion, mothers can safely breastfeed their babies after exposure to iodine contrast agents; they can however stop breastfeeding for 12–24 h if they are still concerned regarding this [18].

### 2.2. Gadolinium-Based Contrast Agents

MRI does not involve radiation: it employs strong magnetic fields, magnetic field gradients, and radio waves to generate images of the organs in the body [19]. MRI during pregnancy is generally considered safe for the fetus, especially in the second or third trimester. The main advantage of MRI compared to US and CT is the ability to visualize the soft tissue structures without the use of ionizing radiation and without depending on an operator’s skills.

Currently, the risks concerning the process of deposition of energy in the body in the form of heat, quantified by the specific absorption ratio (SAR) and measured in units of watts per kilogram (W/kg), are only theoretical.

In animal models, tissue heating caused by elevated SAR during pregnancy, resulting in a rise in maternal body temperature of more than 2–2.5 °C for at least 30–60 min, has been shown to cause fetal harm.

Thus, the FDA and the International Electrotechnical Commission (IEC) recommend not exceeding the maximum SAR of 4 W/kg for the whole body, which is capable of increasing the body temperature by 0.6 °C for 30 min of MRI exposure [19,20].

The main theoretical risks for the fetus exposed to MRI during pregnancy consist of potential teratogenesis, heating of the tissues, which may cause miscarriage and/or injury to organ systems in the first trimester, and acoustic damage resulting from exposure to the magnetic field, but there is currently no evidence of actual damage in animal studies. The main limitations of the studies conducted on humans which are currently available are their retrospective nature and the lack of long-term outcome data [21,22].

According to the guidelines of the American College of Radiology (ACR) and ACOG, MRI, performed with scanners of 3.0T or less, is not associated with any adverse effects on the fetus and no specific precautions are recommended in the first trimester, but it should be used with caution at any gestational age when it is not possible to obtain an accurate diagnosis with other methods and when the information provided may impact on the medical treatment [23,24].

The technique is commonly used in pregnant women to assess acute abdominal and pelvic pain, neurological abnormalities, fetal anomalies (central nervous system, face and neck abnormalities, evaluation of chest, abdominal pelvic masses), localization or abnormally invasive placenta (AIP), myomas, neoplasms, infections, and/or cardiovascular abnormalities.

GBCAs are intravenous drugs used to enhance the quality of MRI or magnetic resonance angiography.

The FDA classifies GBCA as category C agents: their use in animals showed some adverse effects, but studies in humans are lacking. Their use is recommended when the potential benefits overcome the risks as outlined by the American College of Radiology [9].

GBCAs are classified according to their ionicity, the chelating ligand (macrocyclic or linear), their pharmacokinetics, and their risk of causing nephrogenic systemic fibrosis (NSF). The ionic macrocyclic agents represent the safest ones, as they have the highest stability. Contrarily, nonionic linear ones are the least stable, therefore the most dangerous, as Gadolinium in its free form is toxic for humans [25].

The incidence of adverse reactions to GBCAs is low. Most reactions are mild and transient, with skin reactions being the most frequent [26,27].

GBCAs are assumed to cross the placenta in humans since they have been shown to cross the primate’s placenta. These agents accumulate in the amniotic fluid and are not effectively removed from the fetal environment. However, no adverse effects in humans have been reported when clinically recommended doses of GBCAs have been used in pregnant patients. Gadolinium exposure during early pregnancy has been reported in a small number of cases without apparent adverse effects. De Santis et al. investigated if the administration of GBCAs during the first trimester led to teratogenesis or mutagenesis: no effects on newborns were found [28]. Marcos et al. analyzed the impact of exposure to GBCAs during the second and third trimesters of pregnancy in 11 newborns: none of them showed adverse effects at birth [29].

Concerning the breastfeeding period, a small percentage of GBCAs is excreted in breast milk and absorbed by the child [17]. A study revealed that the maximum cumulative amount of gadolinium excreted in breast milk over 24 h was less than 0.04% of the maternal dose and that only 1–2% of such amount was absorbed by the fetus and passed into the bloodstream [30]. There are no reports of adverse effects in newborns breastfed after GBCA administration. Scientific societies agree that it is not necessary to stop breastfeeding, nevertheless, mothers can choose to discard breast milk for 12–24 h after injection of contrast agents as a prevention [31]. 

Long-term risks linked to the GBCA administration include nephrogenic systemic fibrosis (NSF) and retained intracranial gadolinium.

NSF is a rare and severe disease characterized by fibrosing skin lesions and organ failure, observed in patients with impaired renal function. To date, no cases of NSF have been reported in a pregnant patient or newborn after intrauterine exposure [32]. The retained intracranial gadolinium, first described as observed T1 shortening predominantly in the globus pallidus and dentate nucleus, and also observed in patients with normal renal function, has been mainly related to multiple administrations of GBCA during life [21,32].

No evidence is available about the safety during pregnancy of gadoxetic acid, a GBCA used in enhanced MRI to assess liver function useful to detect and characterize liver lesions in patients with known or suspected focal liver disease and to assess the risk stratification of chronic liver disease [33]. As it shows a similar safety profile to other GBCAs for hypersensitivity reactions and NSF but there is incomplete information documenting intracranial gadolinium retention in patients administered gadoxetic acid, its use during pregnancy should be carefully evaluated.

### 2.3. PET CT and Fluorodeoxyglucose

Nuclear studies such as positron emission tomography (PET) are performed by marking a chemical compound with a radioisotope [34].

A PET scan involves the use of positrons and the injection of a radioisotope, fluorodeoxyglucose fluorine 18 (18F-FDG) [35].

The positron combines with an electron and the two opposite charges annihilate each other and produce two gamma rays (annihilation photons) each one 511 keV emitted in opposite directions at approximately 180 degrees from each other [36]. Fluorine 18 (18F) is produced in a cyclotron by bombarding oxygen 18–enriched water with protons and it represents the key principle on which the functioning of the PET scan is based and is administered by intravenous injection [36].

18F-FDG was developed in the 1970s in the first PET center established at the Hospital of the University of Pennsylvania thanks to the collaboration of several researchers such as Martin Reivich, David Kuhl, and Abass Alavi whose contribution led to the first human in history being injected with FDG in August 1976 [37]. No adverse drug reactions linked to 18F-FDG injections that required medical intervention are reported in the official database [36]. Very little information regarding PET scans in pregnancy is available in the literature. Animal reproduction studies have not been conducted with 18F-FDG injections, and it is not known whether a fluorodeoxyglucose F 18 Injection can cause fetal harm when administered to a pregnant patient or if it can affect reproduction capacity.

The FDA classifies 18F-FDG as a category C agent: animal reproduction studies have not been conducted with 18F-FDG injections, so it is not known if it can cause fetal harm when administered to a pregnant woman or if it can affect reproduction capacity. Therefore, 18F-FDG injections should be given to a pregnant woman only if clearly indicated [35].

The main indications to submit a pregnant patient to a PET scan are the need to diagnose a suspected maternal malignancy, to stage an already diagnosticated cancer during pregnancy, or to evaluate the pathophysiology of a brain disorder, such as neurodegenerative disease, infection, epilepsy, seizures, psychiatric affection, and brain tumors [35]. Due to several epidemiological factors, among which is the older woman’s age at first pregnancy, the incidence of oncological diseases during pregnancy is increasing [38].

PET scan maternal side effects are mainly linked to radiation exposure because the injected 18F-FDG seems to be relatively safe when administered in adulthood [35].

18F-FDG crosses the human placenta through two different types of glucose transporters: GLUT1 and GLUT3, as demonstrated by several case reports [39]. Most of the papers available in the present literature have investigated fetus radiation absorption, which seems to be the most dangerous for the health of the growing child. To assess the effect on the stage of fetal growth some works have considered the combination of photon emission from the mother’s body, CT radiation used, and the fetal self-absorption dose [40]. The absorbed dose of 18F-FDG in fetal tissue has been estimated in a case series of 19 pregnant women, but both the short and long-term effects on the fetus remain unclear [41]. The fetal radiation dose from 18F-FDG PET is linked to several factors among which fetal weight, the number of gestation weeks, the maternal administered dose, and the radiotracer used [42,43]. The first trimester of pregnancy is characterized by rapid cell proliferation which is associated with high glucose consumption and, consequently, with a higher fetal absorbed dose [41].

Only a small number of case series about safety data linked to PET scans in human pregnancy have been published, so MRI or CT scans should be preferred to PET when they can provide similar information, but the decision needs to be made on a patient-specific basis.

A recent study about the feasibility and the safety of 18F-FDG PET CT scans in patients with pregnancy-associated cancer by Marie Despierres et al. reported the data of 46 children born from mothers submitted to PET CT scans during pregnancy at 6, 12, 15, and 24 months after birth. No cases of mental retardation, childhood cancer, or malformations were reported by the authors [44]. Similar to other radiopharmaceuticals, 18F-FDG can be transferred to the child during breastfeeding [45].

If the clinical decision is that a PET CT scan should be performed during pregnancy, the mother should have an empty bladder while and a longer imaging time should be used in order to reduce the fetal absorbed dose [40].

The 18F-FDG that is not involved in glucose metabolism in any tissue is then excreted in the urine, so some authors have suggested recurring bladder catheterization in order to maintain an empty bladder and to minimize its concentration in one of the anatomical sites which is closest to the fetus. It is important to assess and verify that the pregnant patient is normoglycemic when undergoing PET imaging with 18F-FDG Injection [46].

As for other radiological exams, low doses are recommended during the first trimester of pregnancy, due to the major fetal sensitivity to ionizing radiation [46].

### 2.4. Barium Sulfate and Gastrographin

Barium agents consist of a suspension of insoluble barium sulfate particles commonly used as the contrast agent of choice to visualize the gastrointestinal tract. Barium agents are not absorbed from the bowel due to their low solubility and cannot be penetrated by x-rays. Barium is contraindicated if bowel perforation is suspected [47]. The major risk is with regard to radiation exposure. In the literature, radiological studies using barium are few in number and most of these studies have been done inadvertently on women during early pregnancy. In fact, barium is not usually performed during pregnancy because of the risk of fetus exposure to ionizing radiation [48].

One of the largest studies was conducted by Han and colleagues. They evaluated 38 women exposed to barium sulfate during early pregnancy and found no association between exposure to barium sulfate and adverse fetal outcomes [48].

Complications and adverse reactions following barium sulfate administration are the same as in the rest of the population.

Gastrographin is a water-soluble contrast media formed by diatrizoate meglumine and diatrizoate sodium which is used as an alternative to barium sulfate in patients who are at higher risk for bowel obstruction or perforation. Its use seems to be safe both orally and by enema solution administration in infants but no evidence is available about its use during pregnancy [49,50].

### 2.5. CEUS

Contrast-enhanced ultrasound during pregnancy has not been approved by leading societies for obstetrics and ultrasound.

CEUS examinations of the upper abdomen sometimes identify focal abnormalities in the liver that may require further investigation, primarily to distinguish liver neoplasia from benign abnormalities.

The utility of CEUS in characterizing focal liver lesions is well-established [51].

The technique of CEUS also results useful in pregnancy to provide an immediate diagnosis in case of an incidental liver lesion.

A retrospective single-center study was performed by Schwarze et al. in 2020. They included six pregnant patients that had undergone CEUS between 2005 and 2014. The applied contrast agent was a second-generation blood-pool agent (SonoVue^®^, Bracco, Milan, Italy).

CEUS was safely performed on all, including pregnant women, without the occurrence of adverse fetal or maternal events [52,53,54].

In a subsequent 2021 study, Schwarze et al. demonstrated the safe and useful application of off-label CEUS in six pregnant women in order to assess different non-obstetric conditions such as desmoid tumor of the abdominal wall, hepatic hemangioma, amebic hepatic abscess, or uncomplicated renal cysts [55,56].

## 3. Discussion

The use of diagnostic imaging has significantly increased in recent years for pregnant or lactating patients for various reasons: the growing desire to obtain a corresponding diagnosis for the given symptomatology, also due to the greater attention regarding medical-legal issues, and the advent of COVID-19 with the symptomatologic set requiring tests to perform a correct differential diagnosis [57,58].

Among the maternal and fetal effects to be considered, in addition to the problems linked to the radiation emitted by the technologies used by modern imaging diagnostics, there are the effects linked to the administration of contrast media.

Among the main fears that mothers have found themselves having to face from 2020 to the present day are those related mainly to hospitalization during the “COVID-19 era”, anti-COVID-19 vaccinations, and the carrying out of a radiological diagnostic examination for maternal health problems.

Regarding COVID-19 vaccination, there is now reassuring evidence regarding vaccinations in pregnancy and fetal protection thanks to the maternal antibody titer [59,60,61]. As far as concerns the toxicity from radiation and contrast media, it appears from the evidence in the literature that the critical dose for fetal health is higher than that of the single radiological examination to which the mother is subjected and that in case of clinical need the radiological examinations currently available can also be performed during pregnancy or breastfeeding [1,8].

Increasing maternal age at first pregnancy and increasing early detection of many malignancies have led to an increase in the diagnosis and treatment of cancers in pregnancy, resulting in an increasing number of diagnostic and prognostic radiological tests [62,63,64].

### 3.1. Medico-Legal Considerations

Missed diagnosis and misdiagnosis represent two of the main causes of medical malpractice claims [31,65,66]. Under both urgent and non-urgent conditions, failure to indicate an imaging examination when it is recommended by guidelines can be used by the patients as evidence of malpractice, unless it is proven that in the specific case the predictable risks for the mother or the fetus related to the technique or to a specific procedure (e.g., administration of contrast agent) outran the benefits. In any case, informed consent is a tool to both defend the gynecologist and empower the patient, and thus it should be recommended that patients be counseled properly before indicating a diagnostic procedure, even when the predictable risks are relatively low [66].

### 3.2. Prevention and Management of Allergic Reactions

Pregnant patients have the same risk factors as the general population of presenting allergic reactions to contrast media. A detailed anamnesis should be performed by clinicians to identify a patient’s risk factors before examination: a previous allergic reaction to contrast agents (5 to 6-fold risk), history of allergies and atopy (3 to 6-fold risk), reactions to other medicaments, older age, history of cardiac or renal disease. A standard oral premedication with prednisone and diphenhydramine should be administrated in patients with a high risk of allergic reactions [18]. Anaphylaxis during pregnancy is a relatively rare but potentially life-threatening event, which requires a prompt diagnosis and treatment to prevent both maternal and fetus morbidity and mortality [67]. Early symptoms of anaphylactic response during pregnancy are vulvar and vaginal itching, low back pain, uterine cramps, fetal distress, and preterm labor [68]. Diphenhydramine is the first-choice treatment in mild to moderate reactions, with prednisone, and dexamethasone for the more severe ones; for acute anaphylaxis, epinephrine and advanced cardiac life support should be provided [69].

### 3.3. Nephrotoxicity

Acute kidney injury (AKI) is considered to be one of the most important complications after intravascular administration of contrast medium, with a lengthening of hospital stays and a consensual increase in costs, morbidity, and mortality. The increase in the frequency of this complication has been explained by the increase in diagnostic imaging investigations that require the administration of contrast medium and also by the fact that it is found more and more frequently in patients who present risk factors for the development of contrast- medium nephropathy [62].

According to the Kidney Disease Improving Global Outcomes (KDIGO) Guidelines, the AKI contrast medium-related is diagnosed by an increase in creatininemia in absolute value ≥ 0.3 mg/dL (44 μmol/L) within 48 h, or relative, of ≥1.5–1.9 from the baseline within 7 days, or a diuresis < 0.5 mL/kg of weight for at least 6 h after exposure to the contrast medium [70].

The pathophysiology of nephropathy after contrast media has been investigated by numerous studies on animals and humans, yet it still remains not fully understood today. In summary, it can be said that the effect of the contrast medium is mediated by two main mechanisms that act synergistically: direct toxicity on renal tubular cells and ischemic damage from tissue hypoxia.

Patients at risk of post-administration contrast medium AKI are united by the presence of risk factors (advanced age, diabetes, renal failure, anemia, cardiovascular disease) that lead to a reduced renal vasodilatory capacity (reduced functional renal reserve), associated with the presence of endothelial dysfunction and atherosclerosis.

In the presence of patients with risk factors for AKI, Creatinine alone does not represent a good index of the patient’s renal function as its value begins to increase significantly only when the GFR is reduced to 50%; the measurement of GFR obtained from creatininemia with the MDRD formula or with the CKD-EPI formula represents the marker of renal function to be used for screening patients’ renal risk.

There are currently no clinical studies reporting differences in the incidence of AKI after radiodiagnostic investigations with the administration of contrast medium in the population of pregnant patients compared to the general population [71].

## 4. Conclusions

The aim of this literature review was to analyze and summarize all the clinical studies available in the literature and Guidelines relating to the use of radiodiagnostic examinations with contrast medium in clinical practice during pregnancy, in order to know and recognize any risks and side effects related to their use and adequately inform women who need radiological examinations during pregnancy.

From the analyzed data it has emerged that the use of contrast media during pregnancy, in particular GBACs, is possible, since they are well tolerated and have a low incidence of adverse effects; however, in clinical practice, it would be better to resort to this type of investigation only when strictly indicated, in order to improve diagnostic accuracy in cases where an investigation without contrast mediums gives limited results and when their use can be lifesaving [72,73,74].

In cases where the use of radiodiagnostic investigations with a contrast medium is necessary during pregnancy, it is therefore essential to discuss its use with patients, improve the advice given regarding the potential benefits that may occur, and provide them an adequate, informed consent.

To date, large prospective studies are actually few in number, so further research should be strengthened to better investigate the risks and benefits of using contrast media in clinical practice and to enable their correct and informed use by physicians.

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
