# Peer review of "Contrast Agents during Pregnancy: Pros and Cons When Really Needed"

_ijerph, 2022, doi:10.3390/ijerph192416699_

Round 1

Reviewer 1 Report

The paper deals with a very interesting topic due to the important implications in clinical practice.

Precisely for this reason it is right that it be published in the most complete and correct way possible.

The paper is not always read fluidly. It deals with a very interesting topic and should be revised in some parts to make it clearer to readers.

- A first major, clearer, distinction must be made between imaging methods that use ionizing and non-ionizing radiation (essential to be considered in pregnancy for any risks to the embryo or fetus) and between the different contrast media to be used (Huynh K. et al. ; Updated guidelines for intravenous contrast use for CT and MRI; Emerg Radiol; 2020. Hasebroock K.M. et al; Toxicity of MRI and CT contrast agents; Expert Opin Drug Metab Toxicol; 2009).

- Line 105: it would be useful to write what the FDA category C indicates ("Animal reproduction studies have shown an adverse effect on the fetus and there are no adequate and well-controlled studies in human beings, but potential benefits may justify the use of the drug in pregnant women despite the potential risks ").

- Line 110: give other information and underline the difference between LOCA and HOCA (especially because LOCA is used more).

- "Gadolinium based contrast agents" (line 152):

 1) "Adverse reactions to GBCAs" (line 195): it is important to consider two of the most severe adverse reactions, Retained Intracranial Gadolinium and Nephrogenic Systemic Fibrosis (Viola L. et al .; MRI in Pregnancy and Precision Medicine: A Review from Literature; J Pers Med; 2022); and the largest human observational study (Ray J.G. et al; Association between MRI exposure during pregnancy and fetal and childhood outcomes; JAMA; 2016).

- "Barium sulfate" (line 280): it is also important to consider (and write) another extravascular contrast agent, Gastrographin.

- "Prevention and management of allergic reactions" (line 347): it is also important to consider another fearful complication, anaphylactic shock, and any therapies to quickly act (Chaudhuri K. et al; Anaphylactic shock in pregnancy: a case study and review of the literature; Int J Obstet Anesth; 2008. Gonzalez-Estrada A. et al; Allergy Medications During Pregnancy; Am J Med Sci; 2016).

- References should be revised: it would be better to insert other useful references to the paper and eliminate others that are not relevant.

Author Response

Dear Reviewer, thank you very much for your constructive comments and suggestions, which had helped us to improve our manuscript. We are very pleased that you found that “the paper deals with a very interesting topic due to the important implications in clinical practice”. We have tried our best to improve and made all the requested changes in the manuscript. A revised manuscript with the correction sections red marked was attached for easy check/editing purpose. Should you have any questions, please contact us without hesitate. Here we have listed the comments and our point-by-point response in detail.

1) A first major, clearer, distinction between imaging methods that use ionizing and non-ionizing
radiation (essential to be considered in pregnancy for any risks to the embryo or fetus) and between
the different contrast media to be used was added in the introduction section, citing the papers
suggested.
2) The FDA category C was reported in its meaning: "Animal reproduction studies have shown an
adverse effect on the fetus and there are no adequate and well-controlled studies in human beings,
but potential benefits may justify the use of the drug in pregnant women despite the potential risks ".
Actual line 132 instead of 105.
3) The structural differences between non-ionic and ionic iodinate contrast media and their relative low
or high osmolality were reported between line 139 and 147 also with the reason why low osmolar
contrast agents are mainly used in the last years.
4) A paragraph about the nephrogenic systemic fibrosis (NSF) and retained intracranial gadolinium
related to GBCA administration was added between line 256 and 271.
5) A paragraph about gastrographin was included between line 362 and line 366.
6) A paragraph about anaphylactic shock and its therapies was included between line 429 and line 437.
7) Please note that English language and style were revised for the second time by an English native teacher.
8) References were updated in order to include papers containing informations useful for the changes required.

Reviewer 2 Report

This is a well-written and detailed review of the safety of contrast materials during pregnancy. 

As seen in the article "https://www.ncbi.nlm.nih.gov/books/NBK507858/#:~:text=Category%20A%3A%20No%20risk%20in,Risk%20cannot%20be%20ruled%20out." FDA offered a new categorization system for pregnancy drug safety in 2015. Updating or adding this categorization would increase the quality of the manuscript. 

-Abstract should contain information about the content of contrast materials summarized in the article, such as CT, MRI, US etc.   -Keywords should be different from the title   -In the CT contrast materials section, the differences about the safety issues between ionic and non-ionic contrast materials should be emphasized.   -Gadolinium based contrast materials:   -Gadoxetic acid can be discussed differently from the others and in more detail.   Discussion:  Similarly, with nephrotoxicity, gadolinium accumulation within different tissues such as the brain can be discussed.

Author Response

Dear reviewer, thank you very much for your constructive comments and suggestions, which had helped us to improve our manuscript. We are very pleased that you found that our study “is well-written and detailed”. We have tried our best to improve and made all the changes required in the manuscript. A revised manuscript with the correction sections red marked was attached for easy check/editing purpose. Should you have any questions, please contact us without hesitate. Here we have listed the comments and our point-by-point response in detail.

1) The FDA category were revised according to the FDA new categorization system for pregnancy drug safety reported in the article cited and the indicated paper was added between the references [7]. 
2) The abstract was modified including the different radiological exams and contrast media analized in the article.
3) Keywords were modified as requested.
4) A paragraph about the safety issues between ionic and non-ionic contrast materials was added between line 78 and 101.
5) Gadoxetic acid was discussed in more detail between line 265 and line 271.
6) GBCA accumulation within different tissues such as the brain was discussed between line 260 and line 264.
7) Please note that English language and style were revised for the second time by an English native teacher.
8) References were updated in order to include papers containing informations to provide the changes required for the revision.

Round 2

Reviewer 1 Report

The paper deals with a very interesting topic, with important implications for clinical practice.

Even if there are some self-citations (62-66) that are not 100% relevant, I appreciate the changes made.